# Do skin bacteriostatic agents reduce acute radiodermatitis in breast cancer patients? A prospective interventional study

Xiaomeng Lu[1], Chao Wei[1], Chuang Li[2], Jiahui Liu[1], Wentong Yang[1], Fang Zhang[1], Zhikun Liu[1], Wenhui Geng[1]*, Yanhong Zhou[1]

1 Radiotherapy department, The Fourth Hospital of Hebei Medical University, Shijiazhuang, China,
2 Department of Operating Room, The Fourth Hospital of Hebei Medical University, Shijiazhuang, China

* gengwenhuidudu73@126.com

## Abstract

### Objective

Radiodermatitis is the most prevalent complication in breast cancer patients undergoing radiotherapy. Emerging evidence suggests a correlation between radiodermatitis and Staphylococcus aureus colonization. The aim of our study was to explore whether the application of skin bacteriostatic agents (a sterile liquid dressing containing fucoidan oligosaccharides with anti-Staphylococcus aureus activity) could mitigate acute radiodermatitis in breast cancer patients.

### Methods

We conducted a prospective interventional study, enrolling breast cancer patients receiving outpatient radiotherapy at our institution from July 1, 2023, to January 31, 2024. Patients were divided into an experimental group and a control group based on the usage of skin bacteriostatic agents. The first outcome was evaluated using the Radiation Therapy Oncology Group (RTOG) criteria. The second outcome was the patient's reported symptoms of radiodermatitis, which included itching, pain, tightness, burning and swelling. The third outcome was whether the usage of skin bacteriostatic agents delayed the onset of self-reported radiodermatitis symptoms.

### Results

A total of 183 patients were enrolled, comprising 101 breast-conserving patients (21 in the experimental group, 80 in the control group); and 82 mastectomy patients (26 in the experimental group, 56 in the control group). The experimental group showed a significantly lower grade of radiodermatitis compared to the control group in both breast-conserving and mastectomy patients ($P < 0.05$). Patient-reported symptoms of radiodermatitis included itching (54.6%), pain (48.1%), tightness (21.9%), burning

**Data availability statement:** The Fourth Hospital of Hebei Medical University owns the data related to this study. The hospital is responsible for holding the data and responding to external data access requests. The contact email address is kycanyi@126.com. All data are stored synchronously at this institution.

**Funding:** This work was funded by the Medical Science Research Project of Hebei in the form of a grant [20150308 to WG].

**Competing interests:** The authors have declared that no competing interests exist.

(11.5%), and swelling (17.5%), but there was no difference between the two groups and onset in radiodermatitis. Further analysis indicated that hypofractionated radiotherapy could reduce the severity of radiodermatitis and itchiness.

## Conclusion

Skin bacteriostatic agents used during radiotherapy can reduce the severity of radiodermatitis in breast cancer patients, although it does not alleviate self-reported symptoms of dermatitis. Hypofractionated radiotherapy is recommended to reduce the severity of radiodermatitis.

## Introduction

According to Global Cancer Statistics 2022, breast cancer is the most prevalent malignancy among women worldwide [1]. A majority of patients undergo adjuvant radiotherapy (RT) following breast-conserving surgery or mastectomy to reduce the risk of local recurrence and enhance overall survival [2]. Radiodermatitis is a common complication, with approximately 30% of breast cancer patients developing grade 2 or higher radiodermatitis [3]. This condition significantly impacts multiple dimensions of quality of life, leading to discomfort, body image disturbances, emotional distress, and interference with daily activities and satisfaction with radiotherapy [4]. In severe cases, radiotherapy may be reluctantly stopped [5].

Despite the high incidence of radiodermatitis, evidence-based treatments remain limited [6]. Both healthcare professionals and patients have been seeking various skin-protective products, such as water-based creams, film dressings, and gels [7–10], to mitigate or avoid radiation damage to normal tissues. A recent study has shown that the skin microbiome in radiation areas changes dynamically with radiodermatitis, in patients with severe radiodermatitis revealed a general non-species-specific overgrowth of skin bacterial load before the onset of severe symptoms. Subsequently, the abundance of commensal bacteria increased in severe radiodermatitis, coinciding with a decline in total bacterial load [11]. Kost's team subsequently identified that Staphylococcus aureus colonization may have something to do with the development of grade 2 or higher acute radiodermatitis in breast cancer patients [12]. The death of skin cells in the radiation field could induce adjacent tissue inflammation and affect the surrounding vascular microenvironment. In other inflammatory skin conditions, such as atopic dermatitis, Staphylococcus aureus has been proven to contribute to pathogenesis [13]. Therefore, from a novel perspective, we hypothesized that applying a skin bacteriostatic agent could potentially reduce the severity of acute radiodermatitis. Jiuer bacteriostatic agent is a liquid dressing refined from various natural herbal antibacterial ingredients, including fucoidan oligosaccharides, which can stimulate the differentiation, maturation, and reproduction of various immunocompetent cells, restore and strengthen the immune system, and exhibit a variety of physiological activities through immune regulation. It is effective in the treatment of chronic skin ulcers and inhibiting opportunistic Staphylococcus aureus in vitro [14].

Currently, the assessment of radiodermatitis is predominantly conducted by healthcare providers, with potential heterogeneity between assessors [15,16]. In clinical practice, clinicians tend to focus solely on treatment outcomes and severe complications [17]. Compared to patients, clinicians significantly underestimate skin symptoms [18], and other common symptoms during radiotherapy, such as hyperpigmentation, dry skin, and pain [17]. Despite the high prevalence of radiodermatitis, less standardized tools are made for patients to assess skin symptoms. Most patient-reported outcome (PRO) tools, which are generic to all skin conditions, require further validation in breast cancer patients [18]. In light of the clinical heterogeneity in the evaluation of radiodermatitis by medical personnel, we selected two outcome indicators. First, the skin toxicity scoring system of RTOG was used by medical staff to assess the grade of radiodermatitis. Second, we included patients' self-reported symptoms of radiodermatitis. We aimed to investigate whether applying skin bacteriostatic agents would delay the onset of self-reported symptoms. Therefore, from the patient's perspective, we explored the items of self-reported symptoms of radiodermatitis.

## Methods and study design

This prospective interventional study enrolled patients who received radiotherapy in the outpatient department from from July 1, 2023, to January 31, 2024. The study protocol was approved by the Ethics Committee of the Fourth Hospital of Hebei Medical University (Registration number: 2023KS117). The eligibility criteria were age ≥ 18 years old; pathologically confirmed diagnosis of breast cancer; no recurrence or metastasis; no active autoimmune disease(s); not use antibiotics or corticosteroid cream during radiotherapy; informed consent and voluntary participation in this study.

### Details of antibacterial agent

The antibacterial agent primarily comprisedβ-D-mannuronic acid (M) andα-L-guluronic acid (G), both of which inhibited the growth of Staphylococcus aureus in skin wounds. The pH value was approximately 6.0.

Manufacturer: Shandong Jiuer Medical Biotechnology Co., Ltd. Net content: 50 ml.

Production record certificate number: Luning Food and Drug Supervision Equipment Production Preparation No. 20150012.

### Design

At the radiotherapy center, all the eligible subjects receiving their first radiotherapy were provided with a radiodermatitis record card (S1 File). Basic information and radiotherapy details associated with radiodermatitis were collected by medical staff. All the patients adhered to general skin care advice, including wearing soft and loose-fitting clothes, avoiding rubbing or scratching irradiated skin, and washing with lukewarm water.

The control group was treated with a medical radiation spray three times a day during radiotherapy. The primary ingredient is superoxide dismutase, which functions by neutralizing oxygen free radicals. These agents were applied over 4 hours before or immediately after radiotherapy. The experimental group was treated with Jiuer bacteriostatic agent, followed by medical radiation spray, three times a day. The medical radiation spray should be applied before the Jiuer bacteriostatic agent on the skin is fully dried out (approximately half an hour). The main component of Jiuer bacteriostatic agent targets the removal of Staphylococcus aureus from the skin surface.

In the first radiotherapy session, nurses instructed patients to apply the agents and ensured their mastery of the technique. Patients were requested to return to the hospital once a week for follow-up visits. Dermatitis was graded by two medical staff using the RTOG skin toxicity scoring system. In cases of disagreement, a third medical staff was consulted until a consensus was reached. Concurrently, patients were inquired about self-reported symptoms of radiodermatitis,which included itching, pain, tightness, burning and swelling. Patients submitted the radiodermatitis record card to medical staff two weeks after the radiotherapy.

## Radiotherapy schedules

All patients underwent radiotherapy in the supine position. The irradiating field for mastectomy patients covered the chest wall±upper and lower clavicular lymphatic drainage areas±inner breast lymphatic drainage area±axillary lymphatic drainage area, with a prescribed dose of 50 Gy. While the irradiating field for breast-conserving patients cloaked the whole breast±tumor-bed, with a whole breast dose of 50 Gy and a tumor-bed boost dose of 6–10 Gy. Conventional fractionated radiotherapy (CRT) of 2 Gy in 25 fractions for a total dose of 50 Gy or hypofractionated radiotherapy (HRT) of 3.1 Gy in 15 fractions for a total dose of 46.50 Gy was administered five times weekly. The treatment duration was 3 weeks for the HRT group and 5 weeks for the CRT group. The radioactive rays were generated by a linear accelerator (Elekta) of 6 Mega electron volts.

## Data collection

A total of 552 records of radiodermatitis were collected from July 1, 2023, to January 31, 2024. Of these, 187 patients were included in the final analysis (Fig 1). General patient characteristics included age, height, weight, history of diabetes mellitus, history of ultraviolet allergy, history of smoking, and type of surgery. Radiotherapy information included the site of radiotherapy and the irradiating dose, which was uniformly converted to an equivalent biological dose for statistical analysis. Two medical staff members collaboratively entered the data into Excel.

## Statistical analysis

According to a literature review, the incidence of grade 2 or higher radiodermatitis is 45%. Using G Power software with α=0.05 (bilateral) and testing the efficacy at the level of 1-β=0.90, the expected incidence of radiodermatitis with skin bacteriostatic

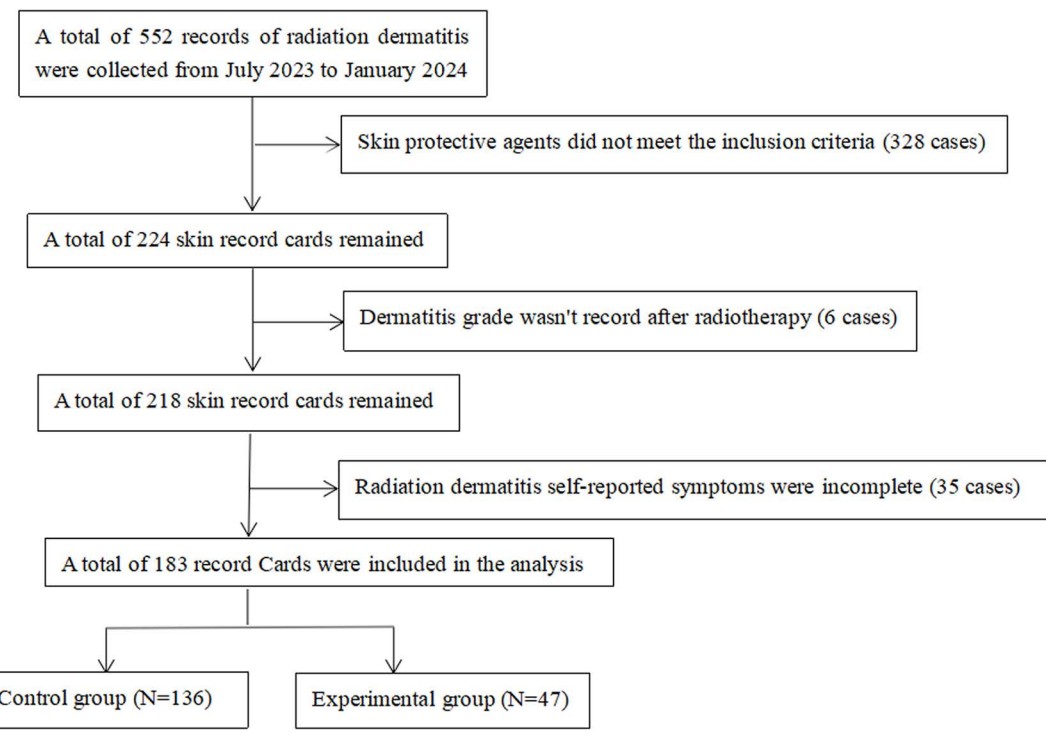

**Fig 1. Study flow diagram.**

agents is 20%. According to the formula for calculating sample size, 41 cases are needed for each of the experimental and control groups. In the real trial, however, 47 cases were collected in the experimental group and 136 in the control group.

Statistical analyses were conducted by SPSS 22. The age distribution of baseline data in mastectomy patients was normally distributed and is presented as mean ± standard deviation; the remainder of the baseline data did not follow a normal distribution and is described as median (quartile) or frequency. Group comparisons were performed using 2-sample t-tests or Wilcoxon rank sum tests or chi-squared tests. Chi-squared and Wilcoxon rank sum tests were utilized to compare the grade of radiodermatitis and self-reported symptoms. Chi-squared tests were employed to assess the impact of different radiotherapy fractionation schedules on radiodermatitis and patient self-reported symptoms. A difference was considered statistically significant at P < 0.05. The Kaplan-Meier curve was plotted to compare the first symptom onset between the two groups and the onset of self-reported symptoms between different radiotherapy methods.

## Results

### Baseline characteristics of the patients

A total of 183 eligible female patients were enrolled in our study, comprising 101 breast-conserving patients (21 patients in the experimental group, 80 in the control group), and 82 mastectomy patients (26 patients in the experimental group, 56 in the control group). None of the patients had a history of smoking (Table 1).

**Table 1. Demographic and clinical characteristics of patients at baseline (n = 183).**

|  | Characteristic | Control group | Experimental group | P [b] |
|---|---|---|---|---|
| Breast-conserving Patients,[a] | Age | 47.50 (42.00,58.00) | 52.00 (44.00,56.00) | 0.58 |
|  | BMI | 24.60 (22.51,26.67) | 26.17 (23.15,28.51) | 0.18 |
|  | Diabetes, n(%) | 8 (10.0) | 4 (19.0) | 0.45 |
|  | Ultraviolet allergy, n(%) | 3 (3.8) | 1 (4.8) | >0.99 |
|  | Radiotherapy dose (breast) | 75.00 (75.00,75.00) | 75.00 (75.00,75.02) | 0.88 |
|  | Breast tumor bed dose | 96.00 (90.34,96.00) | 96.00 (91.45,96.00) | 0.75 |
|  | Radiotherapy dose (supraclavicular), n(%),median(quartile) | 23 (28.8) 75.00 (75.00,75.00) | 8 (38.1) 75.00 (75.00,75.00) | 0.46 |
|  | Radiotherapy dose (internal mammary), n(%),median(quartile) | 8 (10.0) 75.00 (75.00,75.00) | 7 (33.3) 75.00 (75.00,75.00) | >0.99 |
|  | Radiotherapy dose (axillary), n(%),median(quartile) | 10 (12.5) 75.00 (70.31,75.00) | 5 (23.8) 75.00 (74.04,75.00) | 0.53 |
| Mastectomy Patients,[a] | Age,mean(SD) | 50.95(9.03) | 46.85(7.67) | 0.05 |
|  | BMI | 25.39 (23.10,27.65) | 25.80 (23.41,27.93) | 0.93 |
|  | Diabetes, n(%) | 4 (7.1) | 2 (7.7) | >0.99 |
|  | Ultraviolet allergy, n(%) | 1 (1.8) | 1 (3.8) | >0.99 |
|  | Radiotherapy dose (Chest wall) | 75.00 (75.00,75.00) | 75.00 (75.00,75.00) | 0.53 |
|  | Tumour bed dose, n(%),median(quartile) | 2 (3.6) 94.28 (69.42,72.00) | 2 (7.7) 96.00 (72.00,96.00) | 0.32 |
|  | Radiotherapy dose (supraclavicular),median(quartile) | 75.00 (75.00,75.00) | 75.00 (75.00,75.00) | 0.18 |
|  | Radiotherapy dose (internal mammary), n(%),median(quartile) | 39 (69.6) 75.00 (73.08,75.00) | 21 (80.8) 75.00 (75.00,75.00) | 0.12 |
|  | Radiotherapy dose (axillary), n(%),median(quartile) | 10 (17.9) 75.00 (71.60,75.00) | 2 (7.7) 70.13 (65.25,--) | 0.44 |

**Abbreviations:** BMI, body mass index.

aData are presented as median (quartile) of participants unless otherwise indicated.

bUsing a t-test, Chi-square test, and Wilcoxon rank sum tests.

## Radiodermatitis and self-reported symptoms

Using the RTOG criteria, we observed no grade 3 or higher radiodermatitis in either group. The grade of radiodermatitis was significantly lower in the experimental group compared to the control group, with statistically significant differences (Table 2).

In terms of secondary outcomes, self-reported symptoms of radiodermatitis included itching (54.6%), pain (48.1%), tightness (21.9%), burning (11.5%), and swelling (17.5%). However, there were no significant differences between the two groups in self-reported symptoms (Table 3). The time to onset of self-reported symptoms of radiodermatitis did not differ between the two groups of patients (Figs 2–6).

## Comparison of different radiotherapy fractionation schedules

Our study demonstrated that the incidence of grade 2 radiodermatitis and itchiness was lower with hypofractionated radiotherapy than with conventional radiotherapy in breast cancer patients, with statistically significant differences (Table 4). The two fractionation methods did not reduce the time to the first onset of self-reported radiodermatitis symptoms(Fig 7–11).

## Discussions

Our study presents two principal innovations. Firstly, it investigates the potential of skin decontamination to mitigate the severity of radiodermatitis. In addition to assessments by medical personnel, we also evaluated the clinical significance

**Table 2. Comparison the degree of radiodermatitis between the two groups.**

|  | RTOG grades | Control group,n(%) | Experimental group,n(%) | *P* a |
|---|---|---|---|---|
| Breast-conserving Patients | 0 | 3 (3.8) | 4 (19.0) | **0.03** |
|  | 1 | 64 (80.0) | 16 (76.2) |  |
|  | 2 | 13 (16.3) | 1 (4.8) |  |
| Mastectomy Patients | 0 | 1 (1.8) | 0 (0.0) | **0.03** |
|  | 1 | 31 (55.4) | 22 (84.6) |  |
|  | 2 | 24 (42.9) | 4 (15.4) |  |

aUsing Chi-square test.

**Table 3. Self-reported symptoms of radiation dermatitis in both groups.**

|  | Self-reported symptoms | Control group,n(%) | Experimental group,n(%) | *P* a |
|---|---|---|---|---|
| Breast-conserving Patients | Itchiness | 38 (47.5) | 12 (57.1) | 0.43 |
|  | Pain | 50 (62.5) | 10 (47.6) | 0.22 |
|  | Tightness | 8 (10.0) | 4 (19.0) | 0.45[b] |
|  | Burning | 8 (10.0) | 3 (14.3) | 0.87[b] |
|  | Swelling | 18 (22.5) | 4 (19.0) | 0.97[b] |
| Mastectomy Patients | Itchiness | 36 (64.3%) | 14 (53.8) | 0.37 |
|  | Pain | 18 (32.1%) | 10 (38.5) | 0.57 |
|  | Tightness | 22 (39.3%) | 6 (23.1) | 0.15 |
|  | Burning | 7 (12.5%) | 3 (11.5) | >0.99[b] |
|  | Swelling | 8 (14.3%) | 2 (7.7) | 0.63[b] |

aUsing Chi-square test.

bFisher's Exact.

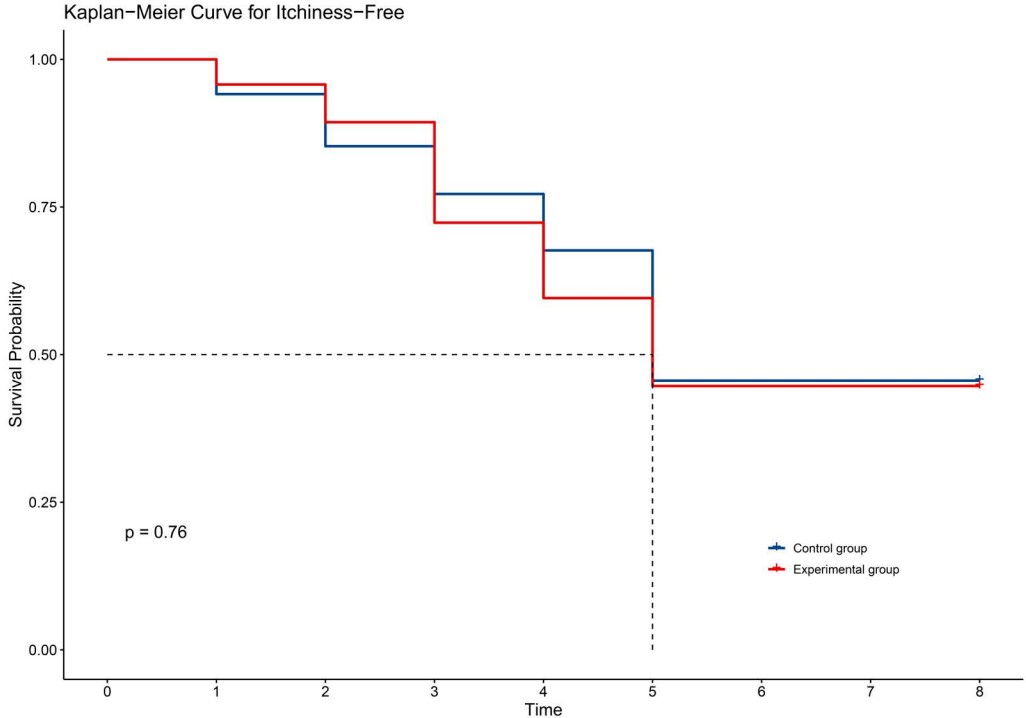

**Fig 2. KM curve for itchiness in two groups.**

of radiodermatitis from the patient's perspective, offering new insights for future research aimed at reducing self-reported symptoms of radiodermatitis.

We evaluated the efficacy of skin protective agents in radiodermatitis from both the clinician and patient perspectives. Our study showed that bacterial decolonization during radiotherapy could reduce the grade of radiodermatitis in breast cancer patients. However, it could not reduce the symptoms of self-reported symptoms or delay the onset time of symptoms of radiodermatitis. In our study, the incidence of grade 2 radiodermatitis was 4.8% in the experimental group of breast-conserving patients, and 15.4% in the mastectomy patients, which was significantly lower than in the control group, consistent with Beth's study [19]. From the patient's perspective, five items were collected for patients self-reported symptoms of radiodermatitis in breast cancer patients, including itchiness, pain, tightness, burning, and swelling, while bacterial decolonization could not reduce self-reported radiodermatitis symptoms or time of first symptom onset. There have been significant advances in radiological techniques and supportive care, however, we still lack effective interventions for the management of radiation-induced skin reactions [6,20]. Previous studies on skin protective agents have been varied and controversial, with the main divide being all skin protection, and the current study explores the idea of skin protection in radiodermatitis from a new perspective. Many skin protectants have been used with the main aim of reducing radiation-induced skin damage, ignoring the dynamics of skin bacteria. The novel idea of this study is to use skin bacteriostatic agents to reduce the severity of radiodermatitis in breast cancer patients.

Radiodermatitis is a common dose-limiting, with intense local inflammatory reactions in the skin after a cumulative radiation dose of more than 10 Gy [21]. The intense inflammatory response can lead to a breakdown in the skin's barrier function, accompanied by bacterial colonization. In 2004, Dr. Madeleine Duvic reported six cases of severe radiodermatitis in cancer patients and drew attention to the role of Staphylococcus aureus in the pathogenesis of severe radiodermatitis [22]. Inflammatory skin diseases such as atopic dermatitis and psoriasis are often colonized by Staphylococcus

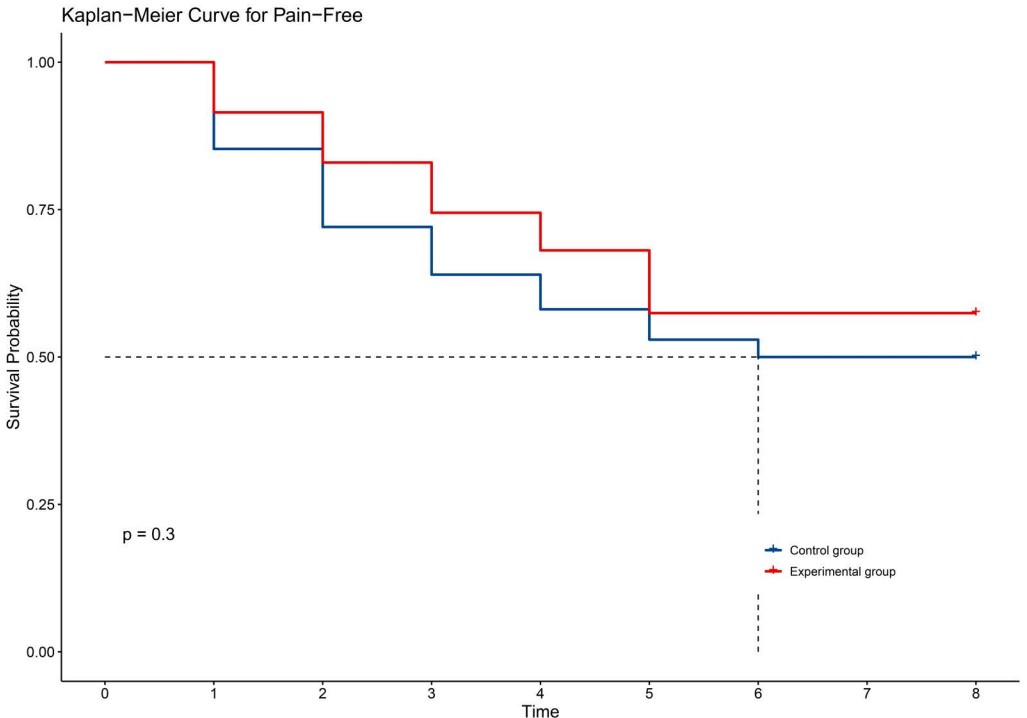

**Fig 3. KM curve for pain in two groups.**

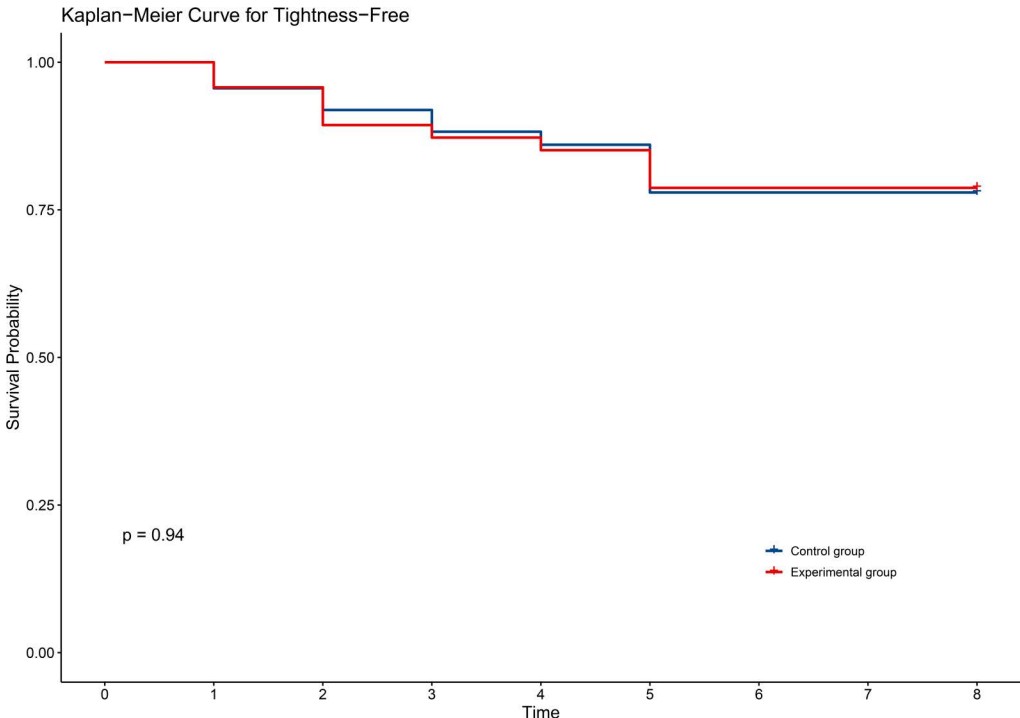

**Fig 4. KM curve for tightness in two groups.**

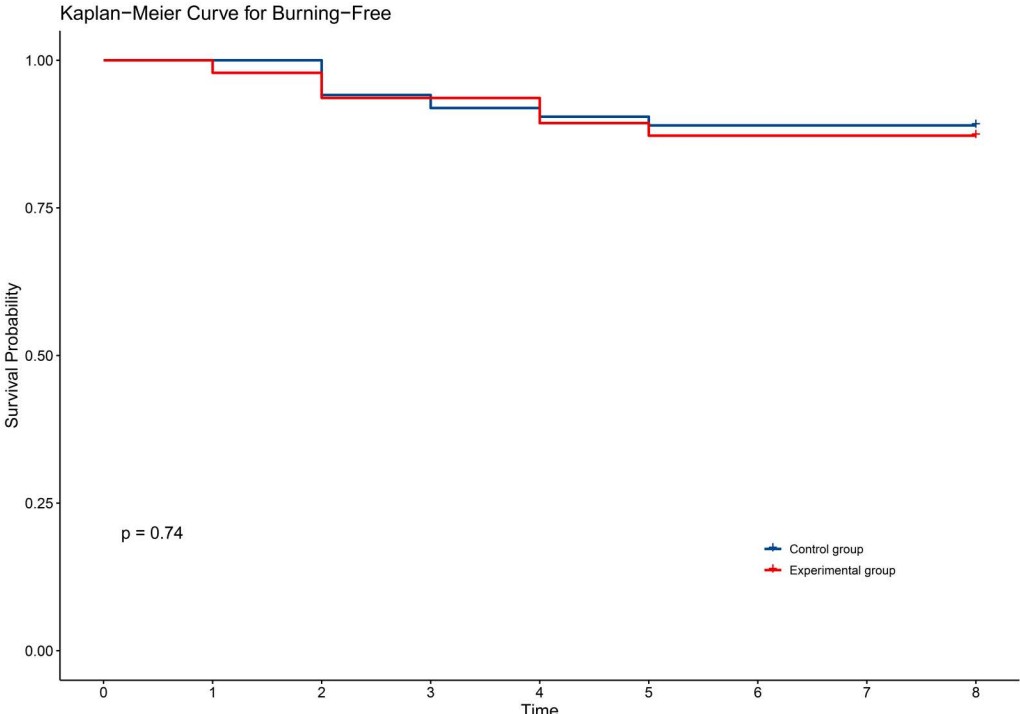

**Fig 5. KM curve for burning in two groups.**

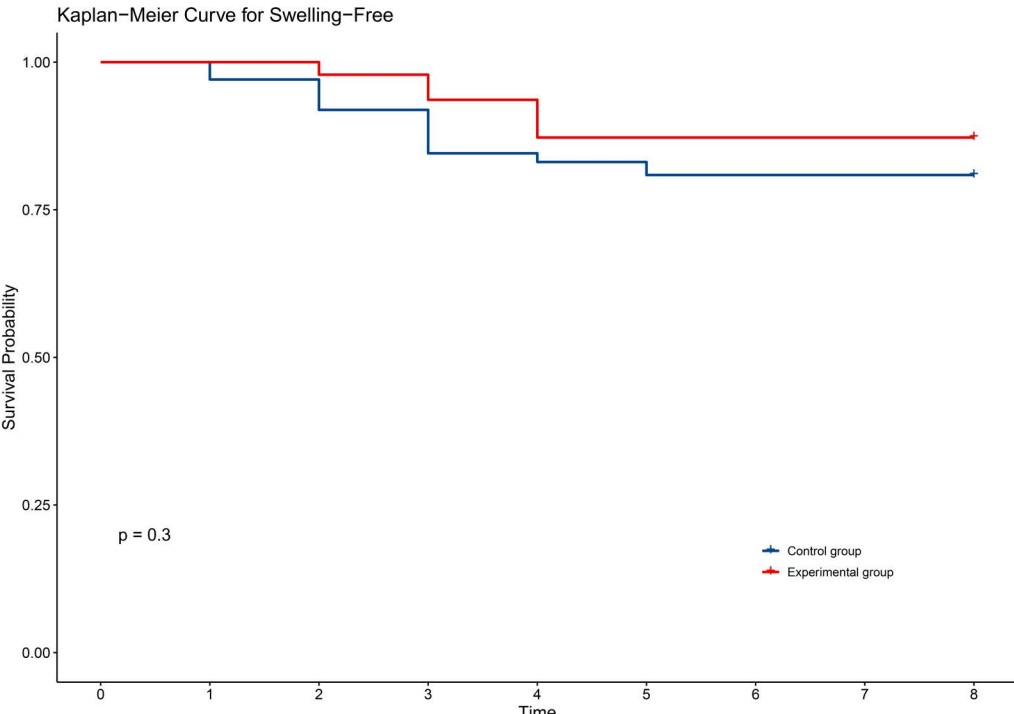

**Fig 6. KM curve for swelling in two groups.**

**Table 4. Comparison of different radiotherapy fractionation schedule for grading radiodermatitis and symptoms associated with radiodermatitis.**

| RTOG grades | Total Patients,n(%) | | | Control group,n(%) | | | Experimental group,n(%) | | |
|---|---|---|---|---|---|---|---|---|---|
| | CRT (n=155) | HRT (n=28) | *P* | CRT (n=112) | HRT (n=24) | *P* | CRT (n=43) | HRT (n=4) | *P* |
| 0 | 4 (2.6%) | 4 (14.3%) | **0.019** | 2 (1.8%) | 2 (8.3%) | 0.190 | 2 (4.7%) | 2 (50.0%) | **0.007** |
| 1 | 114 (73.5%) | 19 (67.9%) | | 78 (69.6%) | 17 (70.8%) | | 36 (83.7%) | 2 (50.0%) | |
| 2 | 37 (23.9%) | 5 (17.9%) | | 32 (28.6%) | 5 (20.8%) | | 5 (11.6%) | 0 (0%) | |
| Self-reported symptoms | | | | | | | | | |
| Itchiness | 90 (58.1%) | 10 (35.7%) | **0.029** | 64 (57.1%) | 10 (41.7%) | 0.167 | 26 (60.5%) | 0 (0%) | 0.072 |
| Pain | 74 (47.7%) | 14 (50.0%) | 0.826 | 56 (50.0%) | 12 (50.0%) | 1.000 | 18 (41.9%) | 2 (50.0%) | 1.000 |
| Tightness | 38 (24.5%) | 2 (7.1%) | **0.041** | 29 (25.9%) | 1 (4.2%) | **0.020** | 9 (20.9%) | 1 (25.0%) | 1.000 |
| Burning | 20 (12.9%) | 1 (3.6%) | 0.270 | 14 (12.5%) | 1 (4.2%) | 0.410 | 6 (14.0%) | 0 (0%) | 1.000 |
| Swelling | 23 (14.8%) | 9 (32.1%) | 0.051 | 17 (15.2%) | 9 (37.5%) | **0.025** | 6 (14.0%) | 0 (0%) | 1.000 |

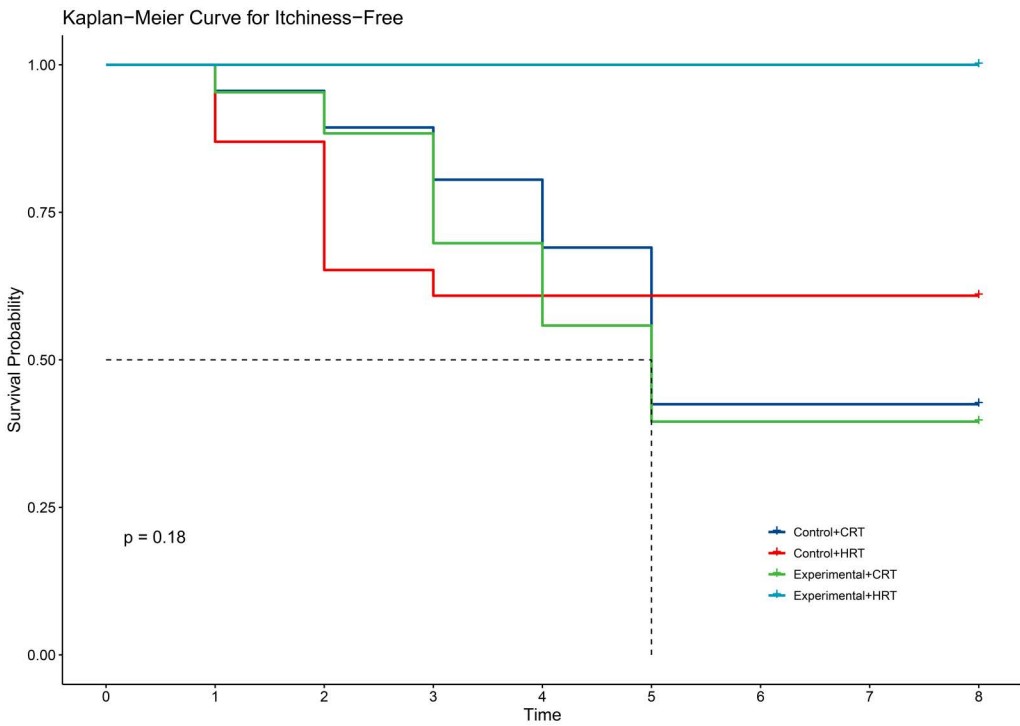

**Fig 7. KM curve for itchiness with different radiotherapy fractionations.**

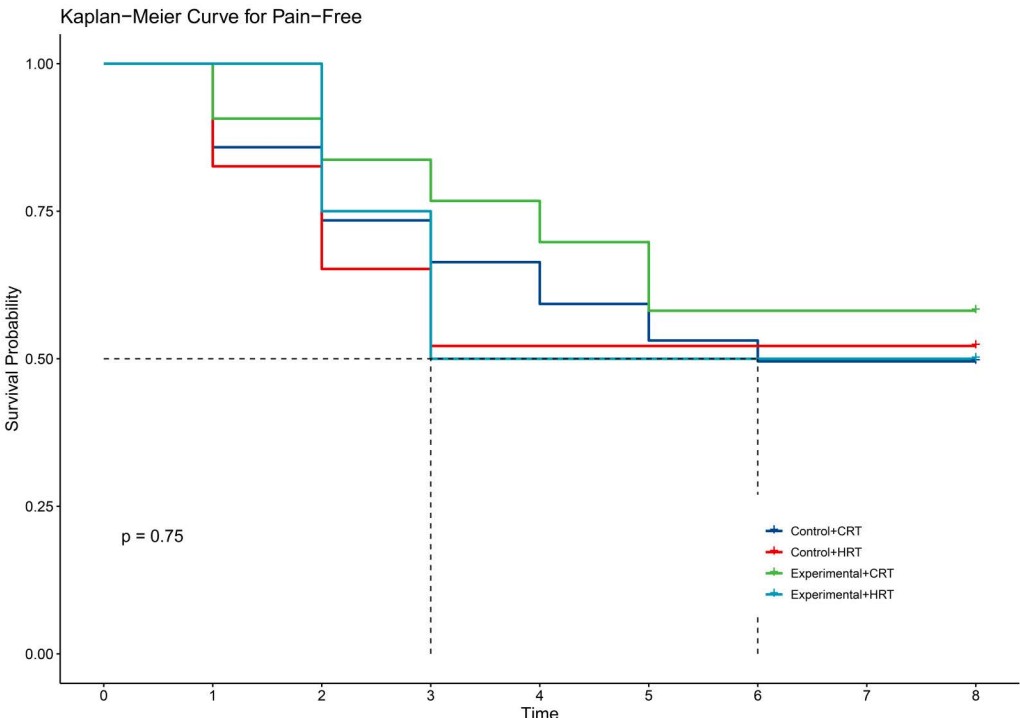

**Fig 8. KM curve for pain with different radiotherapy fractionations.**

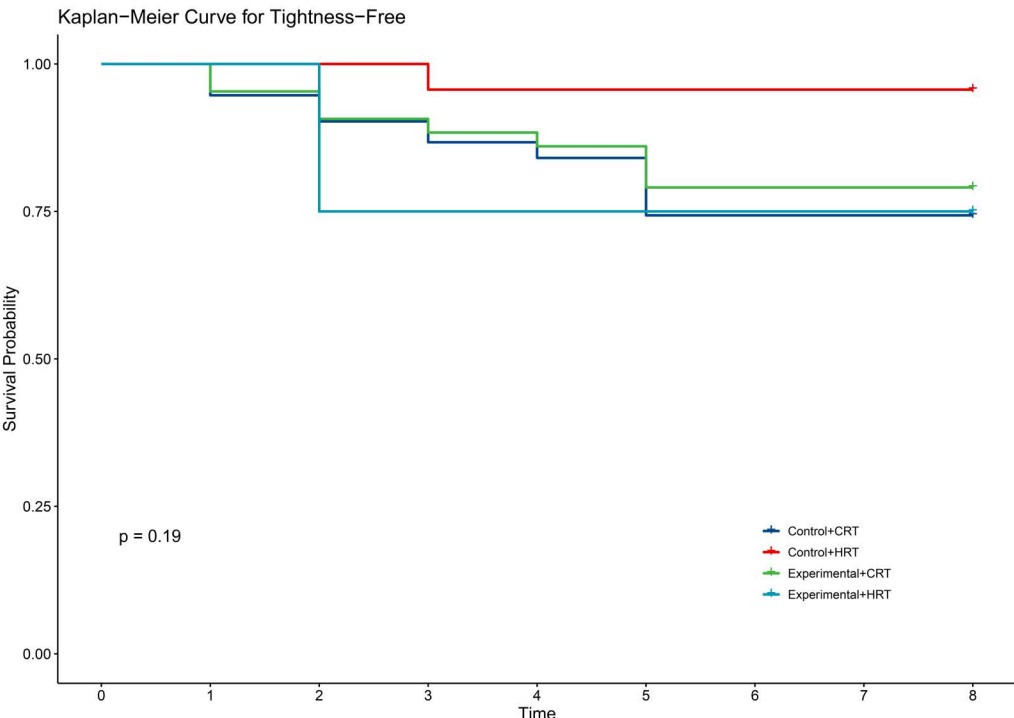

**Fig 9. KM curve for tightness with different radiotherapy fractionations.**

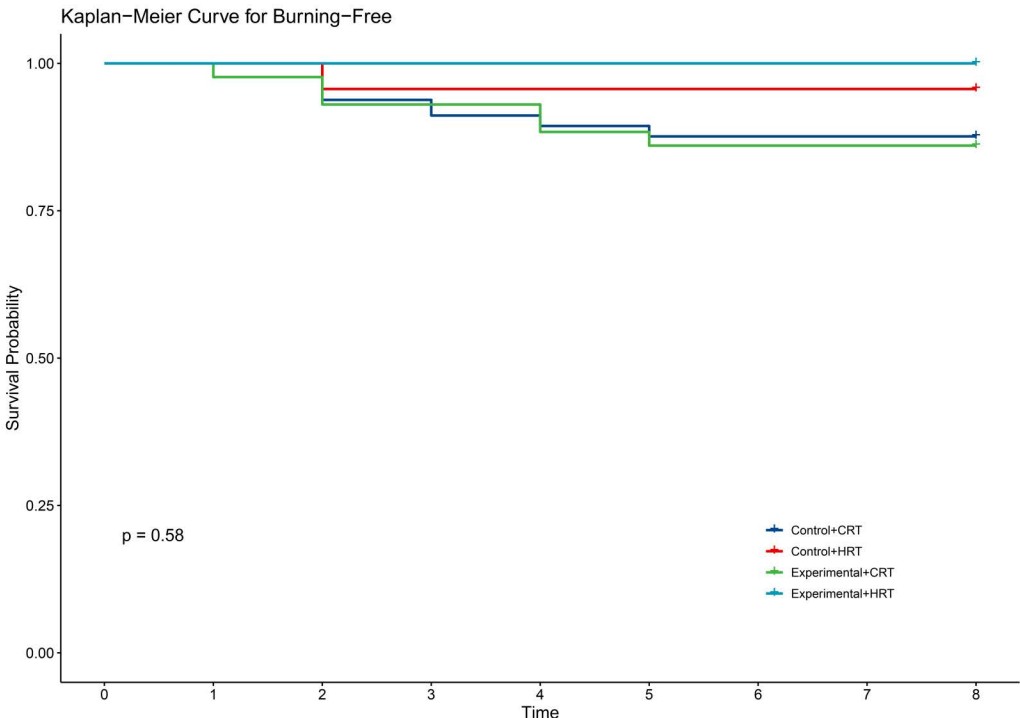

**Fig 10. KM curve for burning with different radiotherapy fractionations.**

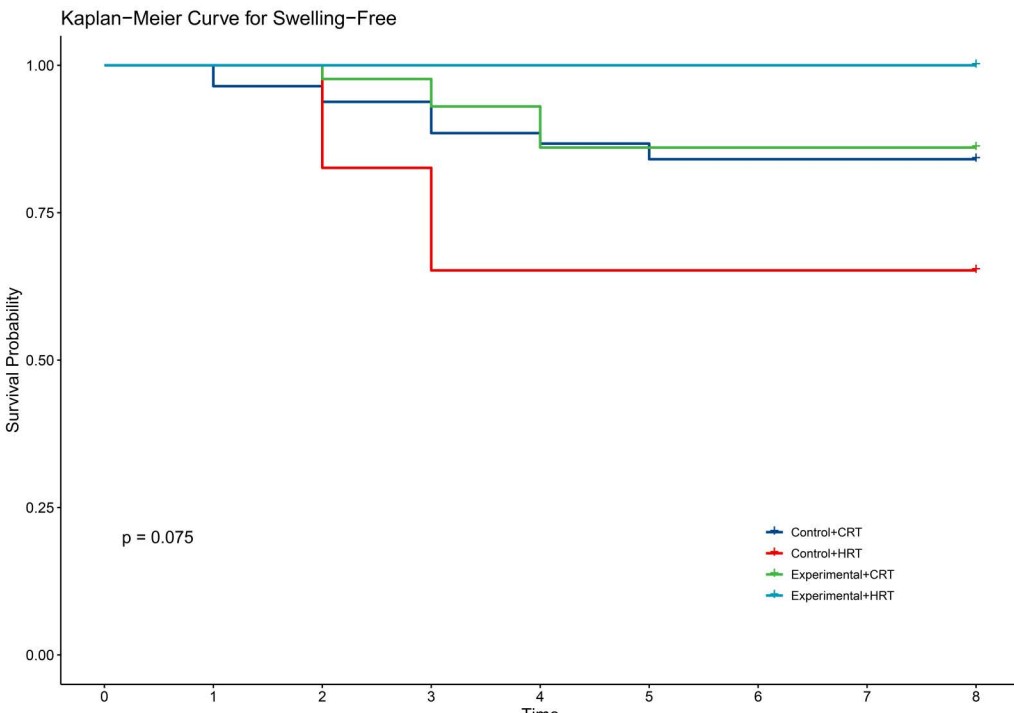

**Fig 11. KM curve for swelling with different radiotherapy fractionations.**

aureus, which releases superantigens, a group of bacterial and viral proteins characterized to stimulate large numbers of T cells. They bind directly to major histocompatibility complex class II molecules on antigen-presenting cells and cross-link antigen-presenting cells with T cells expressing specific T cell receptors, leading to polyclonal T cell activation. When staphylococcal superantigens are applied to intact human skin, clinical manifestations of dermatitis can occur [23]. Thus, bacterial superantigens may exacerbate inflammation by activating T cells and subsequently releasing cytokines, overlapping staphylococcal infections in acute radiodermatitis exacerbate the inflammatory process and impede epidermal barrier repair. Radiodermatitis and atopic dermatitis share the common feature that inflammation leads to disruption of the skin's protective barrier and frequent bacterial colonization, with Staphylococcus aureus isolated from skin lesions in more than 90% of patients with atopic dermatitis and colonizing lesions at rates of up to 107 colony-forming units per square centimeter [24,25]. The bacterium interferes with the inflammatory process in atopic dermatitis in several ways, including the ability to release superantigens in a high proportion of clinical isolates.

Therefore, anti-Staphylococcus aureus therapy is an integral part of treatment for severe atopic dermatitis. In this study, Jiuer Medical Skin Protector, which contains high β-D-(1→4) mannuronic acid M and a-L-(1→4) guluronic acid G chimeric fibers, has an inhibitory effect on Staphylococcus aureus on the skin. Jiuer bacterial decolonization, as a sterile liquid dressing, can change the pathogen's cell membrane, to increase permeability, and rupture the cell, enhancing the organism's immune regulation. Medical radioactive spray is used to prevent and reduce free- radical damage to human skin and mucous membranes caused by the physicochemical factors of radiation. It could not act as an antibacterial agent. At present, few studies on the application of antibacterial agents in radiodermatitis. We look forward to more scholars to conduct further research.

In this study, both patient-reported and provider-reported outcomes were measured, to accurately measure patient experience and provide a reference for the construction of a self-report outcome indicator for radiodermatitis [26]. Additionally, we found five items in the patient self-reported symptoms of radiodermatitis. Among these items, itching and pain were the most common symptoms of radiodermatitis. The pooled analysis of three studies showed that pain in radiodermatitis was common, with a reported rate of 23.3%−83.6% [27–29]. In this study, the reported rate of pain was 47.0%, which was considered to be related to the absence of grade 3 radiodermatitis. The American Pain Society emphasizes understanding variation in patient pain reports from clinical practice so as to facilitate clinical practice change [30]. The 2022 Radiodermatitis Clinical Practice Guideline provides consensus on the use of analgesics to reduce the pain of radiodermatitis and recommends that for grade 2–4 radiodermatitis (CTCAE v4.03), analgesics may improve patient comfort [31]. The next most commonly reported symptom was itchiness. In one study in 2022, the incidence of patient self-reported itchiness was 83.3% [32]. Of 777 patients receiving postoperative radiotherapy for breast cancer, 648 (83.4%) reported itching symptoms, second only to erythema and pain, and 106 (13.6%) reported severe itching that interfered with their life and sleep [33]. However, Jiuer bacterial decolonization did not reduce patients' self-reported symptoms of radiodermatitis and did not delay onset of symptoms. The reason may be that symptoms of radiodermatitis itchiness are associated with serine proteases (e.g. KLK, matriptase, prostaglandins, or trypsin-like enzymes) and single nucleotide polymorphism genotyping (GSTA1 rs3957356-CT, MAT1A rs2282367-GG) [34]. As radiodermatitis progresses, dysfunction of the skin's accessory organs, such as sebaceous glands, sweat glands, and hair follicles, get patients into skin dryness, tingling, tightness, and burning [35]. However, we found that hypofractionated radiotherapy could reduce the severity of radiodermatitis and itchiness, which is consistent with Lu's study [36]. With the progress of radiotherapy technology, more and more patients should be considered for hypofractionated radiotherapy to reduce the side effects of radiotherapy and improve the quality of life. Finally, our relatively small number of cases may reduce the incidence of positive results.

## Conclusions

In this prospective interventional study, the application of skin anti-S. aureus decolonization agent during RT can reduce the severity of radiation dermatitis, but did not reduce self-reported symptoms or delay the onset of symptoms in patients with radiation dermatitis. Hypofractionated radiotherapy is recommended to reduce the degree of radiation dermatitis.

## Limitations

(1) For various reasons, this study did not conduct a rigorous randomized controlled trial. (2) This study was conducted at a single centre, so a multi-centre randomized controlled trial will be needed to validate the results in the future.

## Supporting information

**S1 File. Radiodermatitis record card.**
(PDF)

## Author contributions

Conceptualization: Xiaomeng Lu.

Data curation: Jiahui Liu, Wentong Yang, Fang Zhang.

Formal analysis: Chao Wei.

Investigation: Yanhong Zhou.

Methodology: Chao Wei.

Project administration: Wenhui Geng.

Resources: Wenhui Geng.

Software: Chao Wei.

Validation: Xiaomeng Lu.

Visualization: Chuang Li.

Writing – original draft: Xiaomeng Lu.

Writing – review & editing: Zhikun Liu, Wenhui Geng.

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
