## [Decision Letter · Decision Letter 0]

25 Apr 2025

PONE-D-24-49470Do skin bacteriostatic agents reduce acute radiodermatitis in breast cancer patients? A prospective observational studyPLOS ONE

Dear Dr. Lu,

Thank you for submitting your manuscript to PLOS ONE. After careful consideration, we feel that it has merit but does not fully meet PLOS ONE’s publication criteria as it currently stands. Therefore, we invite you to submit a revised version of the manuscript that addresses the points raised during the review process.

We look forward to receiving your revised manuscript.

Kind regards,

Nukhet Aykin-Burns, PhD

Academic Editor

PLOS ONE

Journal Requirements:

3. In this instance it seems there may be acceptable restrictions in place that prevent the public sharing of your minimal data. However, in line with our goal of ensuring long-term data availability to all interested researchers, PLOS’ Data Policy states that authors cannot be the sole named individuals responsible for ensuring data access (http://journals.plos.org/plosone/s/data-availability#loc-acceptable-data-sharing-methods).

Reviewers' comments:

Reviewer's Responses to Questions

**Comments to the Author**

1. Is the manuscript technically sound, and do the data support the conclusions?

Reviewer #1: Partly

Reviewer #2: Partly

2. Has the statistical analysis been performed appropriately and rigorously? 

Reviewer #1: Yes

Reviewer #2: I Don't Know

3. Have the authors made all data underlying the findings in their manuscript fully available?

Reviewer #1: Yes

Reviewer #2: Yes

4. Is the manuscript presented in an intelligible fashion and written in standard English?

Reviewer #1: Yes

Reviewer #2: Yes

5. Review Comments to the Author

Reviewer #1: This study focused on skin bacteriostatic agents effect on prevention of radiodermatitis in breast cancer patients. Considering high incidence of radiodermatitis in breast cancer patients after radiotherapy, it could be a useful practical study. However, there are some points which should be addressed:

1- In the objectives of the abstract, define which bacteriostatic agents.

2- In the abstract section, you said this is an observational study, but it is interventional. You provide the bacteriostatic agents to some patients and others receive a medical radiation spray.

3- In the abstract method, the first and second outcomes did not write appropriately.

4- Define the abbreviation at first use (e.g. RTOG).

5- In the introduction section, it is better to discuss more comprehensively the logic of the study. Provide more references. Talk more clearly about bacteriostatic formula (Jiuer).

6- Which patient-reporting system did you use for patients’ assessment?

7- Your exclusion criteria are non-including criteria. Edit the title.

8- As you related the effectiveness of the used formula to its antibacterial effect on staphylococcus, use of concomitant antibiotics with anti-staph efficacy could be a confounding factor.

9- History of autoimmune and connective tissue diseases, skin inflammatory diseases (e.g., ectopic dermatitis), concomitant use of nonsteroidal anti‐inflammatory drugs, and other immunosuppressive or antioxidant medications also could be other confounding factors, effective on radiodermatitis incidence. Did you consider them?

10- The main shortage of the method section is the detailed description of the bacteriostatic formula. You should exactly define its ingredients with dose. Where did it prepare or from which company did you buy it?

11- How did you select patients who receive the medical intervention as you did not have randomization?

12- How did you calculate the sample size?

13- How long did you follow the patients weekly? You said that patients were followed weekly but you just assess them in one time point in result section, which is not clear the time of this assessment.

14- The diagram (figure 1) should be edited. You should define how many patients were included in each arm.

15- The limitations of the study are not defined.

Reviewer #2: Radiodermatitis is a prevalent side effect in breast cancer patients undergoing radiation therapy, manifesting as skin irritation that can range from mild redness to severe ulceration. Effective management is crucial to maintain patient comfort and ensure uninterrupted treatment. This is a big problem for cancer patients and agents that will alleviate or prevent the symptoms are not fully effective, thus there is an unmet need to study the phenomena and bring more human data into the literature to develop effective countermeasures.

This study looks at the use and efficacy of a skin bacteriostatic agent to prevent radiodermatitis in cancer patients. This human clinical trial has interesting results and can be an interest to the readers on the radiotherapy field. However, there are certain aspects of the study were not well described in the manuscript and thus the paper needs major revision before it can be accepted for publication.

1-There is no good description of the bacteriostatic agent that was used anywhere in the paper. Is this a wide spectrum?

2-Also the details in the human study like randomization of the patients etc are not described well either.

3-The text needs editing for typos (ex; page 9, “spay” instead of “spray”)

4-What other aspects of the patient histories were considered to eliminate any key factors that could skewed the results. They all need to be at least discussed if considered and the patients were stratified accordingly. If not, it is still needed to be discussed as a limitation of the study.

5-Power calculations during the study design need to be included.

6- Clinical guidelines for radiodermatitis for Oncology Nursing Society provide evidence-based interventions for managing radiodermatitis in cancer patients. It discusses various topical agents, including those with antibacterial properties, for their role in managing skin reactions during radiotherapy and also emphasize individualized patient care and the importance of clinical judgment in selecting appropriate interventions. MASCC skin toxicity group also have guidelines for suggesting the use of various topical agents, with or without antibacterial properties, and their efficacy in managing radiodermatitis. Did either was followed in the study?

6. PLOS authors have the option to publish the peer review history of their article (what does this mean? ). If published, this will include your full peer review and any attached files.

**Do you want your identity to be public for this peer review?** For information about this choice, including consent withdrawal, please see our Privacy Policy .

Reviewer #1: No

Reviewer #2: No

---

## [Author Response · Author response to Decision Letter 1]

29 May 2025

Thank you for giving us the opportunity to submit a revised draft of the manuscript “Do skin bacteriostatic agents reduce acute radiodermatitis in breast cancer patients? A prospective observational study”. We really appreciate the time and effort that you and the reviewers dedicated to providing feedback on our manuscript and are grateful for the insightful comments on and valuable improvements to our paper. We accepted the full comments of the reviewers. Those changes are highlighted within the manuscript. Please see below, for a point-by-point response to the reviewers’ comments and concerns.

Comments from Reviewer 1

1-In the objectives of the abstract, define which bacteriostatic agents.

Answer: Thank you for pointing this out, and we have added this point in the objectives of the abstract.

2-In the abstract section, you said this is an observational study, but it is interventional. You provide the bacteriostatic agents to some patients and others receive a medical radiation spray.

Answer: Thank you for pointing this out and you are very considerate. We have made changes to the study type including the title of the article and the text of the manuscript.

3-In the abstract method, the first and second outcomes did not write appropriately.

Answer: We really appreciate your valuable advice. We have made changes to the manuscript. The first outcome was evaluated using the Radiation Therapy Oncology Group (RTOG) criteria. The second outcome was the patient's reported symptoms of radiodermatitis, which included itching, pain, tightness, burning and swelling.

4-Define the abbreviation at first use (e.g. RTOG).

Answer: We really appreciate your valuable advice. We have made changes in the manuscript.

5- In the introduction section, it is better to discuss more comprehensively the logic of the study. Provide more references. Talk more clearly about bacteriostatic formula (Jiuer).

Answer: We really appreciate your valuable advice. We followed clinical practice guidelines recommending the use of antimicrobial dressings during radiotherapy. In this study, we used the Jiuer antibacterial agent, which is produced by Shandong Jiuer Pharmaceutical Biotechnology Co. Ltd. This agent has an inhibitory effect on Staphylococcus aureus on the skin's surface and fully complies with national requirements. The relevant information about this drug is described in the Methods section.

6-Which patient-reporting system did you use for patients’ assessment?

Answer: Thank you for pointing this out and you are very considerate. A review of the relevant literature revealed that there is no uniform scale for patients to self-report symptoms associated with radiation dermatitis. Therefore, this study used a uniform radiodermatitis record card to observe and record additional symptoms.

7- Your exclusion criteria are non-including criteria. Edit the title.

Answer: We really appreciate your valuable advice. We have amended the inclusion and exclusion criteria in the manuscript.

8- As you related the effectiveness of the used formula to its antibacterial effect on staphylococcus, use of concomitant antibiotics with anti-staph efficacy could be a confounding factor.

Answer: Thank you very much for your valuable advice, your consideration is right. The inclusion criteria for the manuscript have been amended to include patients who were not prescribed antibiotics.

9-History of autoimmune and connective tissue diseases, skin inflammatory diseases (e.g., ectopic dermatitis), concomitant use of nonsteroidal anti‐inflammatory drugs, and other immunosuppressive or antioxidant medications also could be other confounding factors, effective on radiodermatitis incidence. Did you consider them?

Answer: We really appreciate your valuable advice. We have made changes in the inclusion criteria.

10-The main shortage of the method section is the detailed description of the bacteriostatic formula. You should exactly define its ingredients with dose. Where did it prepare or from which company did you buy it?

Answer: Thank you very much for your valuable advice, your consideration is right. We have added details of the skin bacteriostatic agents to the 'Methods' section of the manuscript.

11-How did you select patients who receive the medical intervention as you did not have randomization?

Answer: Thank you for pointing this out and you are very considerate. The application of skin antibacterial agents was based on the skin condition and radiation dose of the patients, as determined by the radiotherapy doctors. Consent was obtained from the patients. Therefore, this study was a non-randomised controlled trial.

12-How did you calculate the sample size?

Answer: We apologize for not mentioning the sample size in our manuscript. According to a literature review, the incidence of grade 2 or higher radiodermatitis is 45%. Using G Power software with α=0.05 (bilateral) and testing the efficacy at the level of 1-β=0.90, the expected incidence of radiodermatitis with skin bacteriostatic agents is 20%. According to the formula for calculating sample size, 41 cases are needed for each of the experimental and control groups. In the real trial, however, 47 cases were collected in the experimental group and 136 in the control group.

13-How long did you follow the patients weekly? You said that patients were followed weekly but you just assess them in one time point in result section, which is not clear the time of this assessment.

Answer: Sorry, we should have made it clearer. This study involves weekly assessments of skin conditions in the radiation fields of patients until two weeks after the end of radiotherapy. The study analyses the most severe cases of radiation dermatitis occurring during radiotherapy and up to two weeks afterwards, as these affect patients more severely.

14-The diagram (figure 1) should be edited. You should define how many patients were included in each arm.

Answer: We really appreciate your valuable advice. We have made changes in figure 1.

15-The limitations of the study are not defined.

Answer: Thank you for pointing this out and you are very considerate. We have added limitations to the manuscript.

Comments from Reviewer 2

1-There is no good description of the bacteriostatic agent that was used anywhere in the paper. Is this a wide spectrum?

Answer: Yes, at present, moderate to severe radiation dermatitis mainly occurs with the intravenous administration of antibiotics to control infection. Topical application of antimicrobial agents to the skin in the radiation field is less common, so our study investigates the topical application of bacteriostatic agents for the treatment of radiation dermatitis, providing a reference point for its prevention.

2-Also the details in the human study like randomization of the patients etc are not described well either.

Answer: Thank you for pointing this out and you are very considerate. This study is a real-world, non-randomised controlled trial, and it has been added to the manuscript.

3-The text needs editing for typos (ex; page 9, “spay” instead of “spray”)

Answer: Thank you for pointing this out, and we have corrected it in the manuscript.

4-What other aspects of the patient histories were considered to eliminate any key factors that could skewed the results. They all need to be at least discussed if considered and the patients were stratified accordingly. If not, it is still needed to be discussed as a limitation of the study.

Answer: We really appreciate your valuable advice. We have made changes in the inclusion criteria.

5-Power calculations during the study design need to be included.

Answer: Thank you very much for suggesting changes to the sample size for this study. We have updated the 'Statistical Analysis' section of the manuscript accordingly.

5-Clinical guidelines for radiodermatitis for Oncology Nursing Society provide evidence-based interventions for managing radiodermatitis in cancer patients. It discusses various topical agents, including those with antibacterial properties, for their role in managing skin reactions during radiotherapy and also emphasize individualized patient care and the importance of clinical judgment in selecting appropriate interventions. MASCC skin toxicity group also have guidelines for suggesting the use of various topical agents, with or without antibacterial properties, and their efficacy in managing radiodermatitis. Did either was followed in the study?

Answer: Yes, we follow clinical guidelines and the MASCC skin toxicity group in our clinical practice. Patients are treated with a radiation spray to protect their skin during radiotherapy, and those with more severe symptoms are given a corticosteroid ointment. In order to minimize confounding factors, patients using corticosteroid ointment were excluded from this study.

---

## [Decision Letter · Decision Letter 1]

3 Jul 2025

Do skin bacteriostatic agents reduce acute radiodermatitis in breast cancer patients? A prospective interventional study

PONE-D-24-49470R1

Dear Dr. Lu,

We’re pleased to inform you that your manuscript has been judged scientifically suitable for publication and will be formally accepted for publication once it meets all outstanding technical requirements.

Kind regards,

Nukhet Aykin-Burns, PhD

Academic Editor

PLOS ONE

Reviewers' comments:

Reviewer's Responses to Questions

**Comments to the Author**

1. If the authors have adequately addressed your comments raised in a previous round of review and you feel that this manuscript is now acceptable for publication, you may indicate that here to bypass the “Comments to the Author” section, enter your conflict of interest statement in the “Confidential to Editor” section, and submit your "Accept" recommendation.

Reviewer #2: All comments have been addressed

2. Is the manuscript technically sound, and do the data support the conclusions?

Reviewer #2: Yes

3. Has the statistical analysis been performed appropriately and rigorously? 

Reviewer #2: Yes

4. Have the authors made all data underlying the findings in their manuscript fully available?

Reviewer #2: Yes

5. Is the manuscript presented in an intelligible fashion and written in standard English?

Reviewer #2: Yes

6. Review Comments to the Author

Reviewer #2: (No Response)

7. PLOS authors have the option to publish the peer review history of their article (what does this mean? ). If published, this will include your full peer review and any attached files.

**Do you want your identity to be public for this peer review?** For information about this choice, including consent withdrawal, please see our Privacy Policy .

Reviewer #2: No

---

## [Editor Report · Acceptance letter]

PONE-D-24-49470R1

PLOS ONE

Dear Dr. Lu,

I'm pleased to inform you that your manuscript has been deemed suitable for publication in PLOS ONE. Congratulations! Your manuscript is now being handed over to our production team.

Kind regards,

on behalf of

Dr. Nukhet Aykin-Burns

Academic Editor

PLOS ONE